# Long-Term Systemic Expression of a Novel PD-1 Blocking Nanobody from an AAV Vector Provides Antitumor Activity without Toxicity

**DOI:** 10.3390/biomedicines8120562

**Published:** 2020-12-02

**Authors:** Noelia Silva-Pilipich, Eva Martisova, María Cristina Ballesteros-Briones, Sandra Hervas-Stubbs, Noelia Casares, Gualberto González-Sapienza, Cristian Smerdou, Lucia Vanrell

**Affiliations:** 1Division of Gene Therapy and Regulation of Gene Expression, Cima Universidad de Navarra, 31008 Pamplona, Spain; nsilva.1@alumni.unav.es (N.S.-P.); emartisova@alumni.unav.es (E.M.); mballestero.3@alumni.unav.es (M.C.B.-B.); 2Instituto de Investigación Sanitaria de Navarra (IdISNA), 31008 Pamplona, Spain; mshervas@unav.es (S.H.-S.); ncasares@unav.es (N.C.); 3Cátedra de Inmunología, DEPBIO, Facultad de Química, Instituto de Higiene, UDELAR, 11600 Montevideo, Uruguay; ggg2727@gmail.com; 4Division of Immunology and Immunotherapy, Cima Universidad de Navarra, 31008 Pamplona, Spain; 5CIBERehd, Instituto de Salud Carlos III, 28029 Madrid, Spain; 6Facultad de Ingeniería, Universidad ORT, 11100 Montevideo, Uruguay

**Keywords:** nanobody, VHH, PD-1, PD-L1, AAV vector, cancer, gene therapy, immunotherapy

## Abstract

Immune checkpoint blockade using monoclonal antibodies (mAbs) able to block programmed death-1 (PD-1)/PD-L1 axis represents a promising treatment for cancer. However, it requires repetitive systemic administration of high mAbs doses, often leading to adverse effects. We generated a novel nanobody against PD-1 (Nb11) able to block PD-1/PD-L1 interaction for both mouse and human molecules. Nb11 was cloned into an adeno-associated virus (AAV) vector downstream of four different promoters (CMV, CAG, EF1α, and SFFV) and its expression was analyzed in cells from rodent (BHK) and human origin (Huh-7). Nb11 was expressed at high levels in vitro reaching 2–20 micrograms/mL with all promoters, except SFFV, which showed lower levels. Nb11 in vivo expression was evaluated in C57BL/6 mice after intravenous administration of AAV8 vectors. Nb11 serum levels increased steadily along time, reaching 1–3 microgram/mL two months post-treatment with the vector having the CAG promoter (AAV-CAG-Nb11), without evidence of toxicity. To test the antitumor potential of this vector, mice that received AAV-CAG-Nb11, or saline as control, were challenged with colon adenocarcinoma cells (MC38). AAV-CAG-Nb11 treatment prevented tumor formation in 30% of mice, significantly increasing survival. These data suggest that continuous expression of immunomodulatory nanobodies from long-term expression vectors could have antitumor effects with low toxicity.

## 1. Introduction

Immunotherapies based on the use of monoclonal antibodies (mAbs) against immune checkpoints have shown great promise in treating many cancer types, revolutionizing the field of cancer therapy [1,2]. One of the immune checkpoints that many tumors exploit to evade immune eradication is the programmed death-1 (PD-1)/PD-L1 (programmed death ligand 1) axis, which functions as a major negative immune regulator [3]. PD-1 is mainly expressed on T cells upon activation, whereas its main ligand, PD-L1, is expressed on the surface of a variety of cells (including T and B cells, natural killer cells, antigen presenting cells, epithelial cells, and vascular endothelial cells), and it is further upregulated in inflammatory contexts [4,5]. Likewise, PD-L1 can be expressed by different types of tumor and tumor-associated cells, and even by tumor-derived extracellular vesicles, contributing to the evasion of antitumor immune responses [6,7,8,9,10]. Altogether, the PD-1/PD-L1 pathway is considered an interesting target for immunotherapy, and several mAbs blocking this interaction have already been approved for clinical use, including anti-PD-1 pembrolizumab and nivolumab, and anti-PD-L1 atezolizumab, avelumab, and durvalumab [11,12]. Although blockade of the PD-1/PD-L1 pathway with mAbs has elicited durable clinical responses in patients with a wide array of malignancies, it is undeniable that the efficacy of these treatments needs improvement. Moreover, immunomodulatory mAbs administered intravenously frequently cause immune-related adverse events [13,14,15].

One of the main disadvantages of conventional mAbs is their limited tumor penetration due to their large size (~150 kDa). Nanobodies (Nbs, also referred to as VHHs [variable fragments of heavy chain antibodies] or single-domain antibodies) derive from the variable fragments of heavy-chain only antibodies (HcAbs) present in the *Camelidae* family [16]. They are the smallest naturally derived antigen-binding domains known so far (~15 kDa), and present several advantages including high solubility, stability, specificity, affinity, as well as the possibility to be efficiently expressed from both prokaryotic and eukaryotic systems. Furthermore, Nbs can be easily engineered to generate multi-domain constructs or to be conjugated with additional proteins, reporter molecules, or drugs [17]. Moreover, protocols for Nb generation from peripheral blood cells of immunized camelids using the phage display technique are well established [18]. Due to their unique properties, Nbs have made their way into both biomedical research and clinical applications. In fact, several Nbs are currently being evaluated in clinical trials for treatment of different diseases, and a Nb has been recently approved for the treatment of thrombotic thrombocytopenic purpura disease [19,20]. In the field of cancer therapy, Nbs are very attractive tools because they show high extravasation and tissue penetration (including tumors) and low immunogenicity. However, their small size might also be a handicap, because they are rapidly eliminated from the bloodstream through renal clearance [20,21].

One strategy to overcome this potential drawback could be the use of gene therapy vectors able to express Nbs in vivo at therapeutic levels for a prolonged time. Of note, Nbs are good candidates for gene therapy, due to their small size and efficient expression in mammalian cells [22]. Among the different gene therapy vectors, adeno-associated virus (AAV) has been frequently used in clinical development. AAV vectors can provide long-term gene expression after delivery without genomic integration, they are non-pathogenic, and they present a good safety profile. Administration of recombinant AAV (rAAV) has been shown to be safe and well-tolerated, and treatments based on rAAVs have already been approved for clinical use [23]. For cancer immunotherapy, a single vector administration could potentially achieve sustained expression of the immunomodulatory agent, avoiding the need for repetitive injections of high antibody doses. As a result, mAb delivery with AAV could become more affordable and safer.

In the present study, we selected and characterized a novel anti-PD-1 Nb that was able to inhibit the PD-1/PD-L1 interaction of both mouse and human molecules In vitro. We generated AAV vectors that were able to provide high and sustained Nb expression in mice without apparent toxicity. Significant protection against a tumor challenge with colon adenocarcinoma cells was observed in these mice compared to the control group.

## 2. Materials and Methods

### 2.1. Cell Lines and Animals

BHK cells (ATCC-CCL10) were cultured in GMEM-BHK21 (Thermo Fisher, Waltham, MA, USA) supplemented with 5% fetal bovine serum (FBS), 10% tryptose phosphate broth, 2 mM glutamine, 20 mM HEPES, and antibiotics (100 µg/mL streptomycin and 100 U/mL penicillin). HEK-293T (ATCC-CRL-3216) and HuH-7 (Japanese Collection of Research BioResources Cell Bank: 0403) cells were cultured in DMEM (Thermo Fisher, Waltham, MA, USA) supplemented with 10% FBS, 2 mM glutamine, 100 μg/mL streptomycin and 100 U/mL penicillin. MC38 cells were a kind gift from Dr. Karl E. Hellström (University of Washington, Seattle, WA, USA) and were cultured in RPMI-1640 medium (Lonza, Basel, Switzerland) supplemented with 10% FBS, 2 mM glutamine, 20 mM HEPES, antibiotics, and 50 µM 2-mercaptoethanol.

Four-week-old female C57BL/6 mice were purchased from Envigo (Barcelona, Spain). Animal studies were approved by the Universidad de Navarra ethical committee (study number 024-18) for animal experimentation under Spanish regulations. An adult female llama (*Lama glama*) from Montevideo municipal zoo (Montevideo, Uruguay) was used for immunization and Nb library construction. The protocol was approved by the Parque Lecocq ethical committee and manipulation of the llama was performed by veterinarians.

### 2.2. Llama Immunization and Library Construction

The plasmid for recombinant expression of mouse PD-1 (mPD-1) ectodomain was kindly provided by Dr. Steven C. Almo (Albert Einstein College of Medicine, New York, NY, USA). Expression of mPD-1 ectodomain was performed as described elsewhere [24], and purity was evaluated by SDS-PAGE. The purified mPD-1 ectodomain was used for the llama immunization in incomplete Freund adjuvant by subcutaneous injection (4 doses of 600 μg/each given every three weeks). One week after the last boost, 200 mL of blood were collected from the femoral artery in a collection bag with 50 mL of anticoagulant solution (25 mM citric acid, 51 mM sodium citrate and 74 mM dextrose), and used for isolation of peripheral blood mononuclear cells with Histopaque-1077 gradients (Sigma, St Louis, MO, USA), according to the manufacturer’s instructions. Total RNA was extracted from 5 × 10^7^ cells with TRIZOL reagent (Invitrogen, Waltham, MA, USA), quantified spectrophotometrically and used for cDNA synthesis using the superscript III first-strand synthesis system for RT-PCR (Invitrogen, Waltham, MA, USA) and oligo dT. DNA fragments encoding the heavy chain variable domains of conventional and heavy chain only antibodies (VH and VHH, respectively) were amplified by PCR using a mix of three different forward primers (VH1: 5′-CATGCCATGACTCGCGGCCCAGGCGGCCATGGCCCAGGTGCAGCTGGTGCAGTCTGG-3′, VH3: 5′-CATGCCATGACTCGCGGCCCAGGCGGCCATGGCCGAGGTGCAGCTGGTGGAGTCTGG-3′ and VH4: 5′- CATGCCATGACTCGCGGCCCAGGCGGCCATGGCCCAGGTGCAGCTGCAGGAGTCGGG-3′), and one reverse primer (JH: 5′-CCACGATTCTGGCCGGCCTGGCCTGAGGAGACRGTGACCTGGGTCC-3′), as described previously [25]. These fragments were cloned into the pComb3X phagemid vector (Addgene, Watertown, MA, USA), a kind gift from Dr. Carlos F. Barbas (The Scripps Research Institute, La Jolla, CA, USA). VHHs fragments (Nbs) cloned into this vector are expressed fused to a 6 × His tag, a hemagglutinin (HA) tag and the *pIII* gene at their carboxy-terminal end. The ligation was electroporated into competent *Escherichia coli* ER2738 cells (Lucigen Corporation, Middleton, WI, USA), which were subsequently infected with M13KO7 helper phage (Pharmacia Biotech, Solna, Sweden) to induce the production of phages expressing VH fragments and Nbs, as described previously [26].

### 2.3. Selection of Specific Nanobodies

For the selection of specific Nbs against PD-1, we used HEK-293T cells expressing mPD-1 or human PD-1 (hPD-1). To generate these cells, the genes coding for the entire sequence of mPD-1 and hPD-1 were synthetized by Integrated DNA Technologies (Coralville, IA, USA), based on sequences available in GenBank (NM_008798 and NM_005018, respectively), and subcloned into the pEF1/Myc-His vector (Invitrogen, Carlsbad, CA, USA) for expression in eukaryotic cells. The generated plasmids were used to transfect HEK-293T cells using linear polyethylenimine (PEI) of 25 kDa (Polysciences, Warrington, PA, USA) as transfection agent. Transfection conditions were first optimized using a plasmid coding for green fluorescent protein (GFP) and evaluating cells by flow cytometry. 48 h post-transfection, cells were harvested and used immediately or stored at −80 °C in FBS with 10% dimethyl sulfoxide (DMSO) for later use. PD-1 expression in transfected cells was confirmed by flow cytometry using anti-PD-1 antibodies fused to phycoerythrin (eBioscience, San Diego, CA, USA).

For selection of anti-PD-1 Nbs, 100 µL of the phage library was pre-incubated with 100 µL of wild type HEK-293T cells (10^7^ cells/mL in FACS [flow cytometry staining buffer] buffer: PBS supplemented with 1% bovine seroalbumin (BSA), 25 mM HEPES and 0.05% sodium azide) for 1 h at 4 °C. After that, cells were centrifuged and the supernatant (containing the unbound phages) was incubated with 100 µL of HEK-293T cells expressing mPD-1 (10^7^ cells/mL in FACS buffer) for 1 h at 4 °C. Cells were washed four times with 1 mL of FACS buffer, pelleted, and incubated with 150 μL of 10 mg/mL trypsin for 30 min at 37 °C to recover cell surface bound phages. After centrifuging at high speed for 5 min, the supernatant containing the eluted phages (output) was collected and used for titration and amplification. For the titration, ten-fold serial dilutions of the output were used to infect 100 µL of an ER2738 *E. coli* culture grown until OD_600nm_ reached 0.8 absorbance units (AU), for 30 min at 37 °C. After that, cells were seeded in LB-ampicillin plates and incubated overnight (o/n) at 37 °C. The next day, output titers were calculated by counting individual colonies in the appropriate dilution and expressed as colony forming units (cfu)/mL. The rest of the output was used for amplification, for which 2 mL of ER2738 *E. coli* at OD_600nm_ of 0.8 AU were infected with the output and subsequently over-infected with M13KO7 helper phage. Production of phages was carried out o/n at 37 °C with shaking in 12 mL of Super Broth (SB) medium supplemented with ampicillin and kanamycin, and the next day, cells were pelleted and the supernatant (containing the phages) was incubated with 2.5 mL of a 20% PEG8000 and 2.5 M NaCl solution, for 1 h at 4 °C. Phages were pelleted by centrifugation at 20,000× *g* for 20 min at 4 °C and resuspended with 1 mL of PBS–3% BSA supplemented with protease inhibitor cocktail (Roche, Basel, Switzerland). The amplified output was used for the next round of selection. Three rounds were performed in total. This process allowed us to select phages displaying Nbs able to bind mPD-1, while phages displaying VHs were most likely not selected since in that case the light variable chain would also be needed for antigen recognition.

### 2.4. Screening for Specific Nanobodies

After the final round of selection, individual colonies from the titration plates were randomly selected and cultured in 96 deep-well blocks in LB-ampicillin. Expression of Nbs was induced with 1 mM IPTG when OD_600nm_ reached 0.6 AU. Cultures were incubated o/n at 37 °C with shaking, after which supernatants (containing soluble Nbs) were collected and used for screening of specific clones against mPD-1 and hPD-1 by ELISA and flow cytometry.

For ELISA, 96-wells plates were coated with hPD-1-Fc or mPD-1-Fc (R&D Systems, Minneapolis, MN, USA) at 1 µg/mL in PBS o/n at 4 °C. Wells were blocked for 1 h at room temperature (RT) with 1% BSA in PBS, washed with PBS–0.05% Tween 20 (PBST), and incubated with supernatants containing Nbs diluted in PBST–0.2% BSA for 1 h at RT. Nbs were detected with a rat anti-HA antibody conjugated to peroxidase (Roche, Basel, Switzerland) for 1 h at RT. The assay was developed by addition of 100 μL/well of tetramethylbenzidine (TMB) substrate solution (BD Biosciences, Franklin Lakes, NJ, USA). After 15 min, the reaction was stopped by the addition of 50 μL/well of 2 N H_2_SO_4_, and the absorbance was read at 450 nm on a FLUOstar Optima Reader (BMG, Berlin, Germany).

For flow cytometry, wild type HEK-293T cells or transiently transfected HEK-293T cells expressing mPD-1 or hPD-1 were seeded in V-bottom 96-wells plates (2.5 × 10^5^ cells/well in 50 µL of FACS buffer) and incubated with supernatants containing Nbs (50 µL/well) for 1 h at 4 °C. After washing with FACS buffer, cells were stained with an anti-HA antibody conjugated to phycoerythrin (Abcam, Cambridge, UK) for 1 h at 4 °C, washed, and analyzed by flow cytometry in a FACSCalibur (BD Biosciences, Franklin Lakes, NJ, USA) followed by analysis with FlowJo (TreeStar, Ashland, OR, USA).

### 2.5. Large-Scale Production of Nanobodies

For expression in *E. coli*, selected Nbs were subcloned into the pINQ-H6HA vector between two different Sfi I restriction sites as described previously [25]. Nbs expressed from this vector are fused to a HA tag and a 6 × His tag in the carboxy-terminal end. Ligations were transformed into BL21 (DE3) *E. coli* and grown in LB-kanamycin. Nb expression was induced with 3 µM IPTG for 4 h at 37 °C with agitation, after which cells were pelleted and periplasmic proteins were extracted by sonication. Nbs were purified using Ni-NTA columns in an ÄKTA purification system (GE Healthcare, Chicago, IL, USA) according to the manufacturer’s instructions. For cell culture, purified Nbs were further treated with High-Capacity Endotoxin Removal Resin columns (Thermo Fisher) and absence of endotoxins was verified with the Kinetic-QCL^TM^ Kinetic Chromogenic LAL Assay (Lonza, Basel, Switzerland).

### 2.6. PD-1/PD-L1 Binding ELISA

The ability of anti-PD-1 Nbs to inhibit PD-1/PD-L1 binding was evaluated by ELISA. For human variants, 96-wells plates were coated with 100 µL/well of hPD-1-Fc (R&D Systems, Minneapolis, MN, USA) at 1 µg/mL in PBS o/n at 4 °C, blocked with PBS–0.5% BSA for 1 h at RT, washed with PBST, and incubated with biotinylated hPD-L1-Fc (BPS Bioscience, San Diego, CA, USA) at 0.2 µg/mL in PBST–0.2% BSA for 2 h at RT. PD-1/PD-L1 binding was detected with streptavidin conjugated to horse radish peroxidase and subsequent incubation with TMB substrate solution for 15 min. For the murine variants, plates were coated with 1 µg/mL of mPD-L1-Fc (R&D Systems, Minneapolis, MN, USA), blocked and washed as described before, and incubated with biotinylated mPD-1-Fc (BPS Bioscience, San Diego, CA, USA) at 0.25 µg/mL. Selected Nbs were incubated at different concentrations (from 30 to 0.3 nM approximately), simultaneously with biotinylated mPD-1-Fc or hPD-L1-Fc, to evaluate their ability to inhibit binding of PD-1 to PD-L1.

### 2.7. Mouse T Cell Reactivation In Vitro

Mouse T cells were purified from C57BL/6 mice spleens using the Pan T cell isolation kit and the autoMACS Pro Separator (Miltenyi Biotec, Bergisch Gladbach, Germany), according to the manufacturer’s instructions. U-bottom 96-wells plates were coated with 200 µL/well of a mix of anti-mCD3 antibody (eBioscience, San Diego, CA, USA) and mPD-L1-Fc at 5 µg/mL each in PBS o/n at 4 °C. After washing the plate with PBS, wells were incubated with purified T cells (10^5^ cells/well), anti-mCD28 antibody (eBioscience, San Diego, CA, USA) at a final concentration of 1 µg/mL, and anti-PD-1 Nb or control antibodies, in a final volume of 250 µL RPMI/well. After 72 h, supernatants were collected and levels of murine interleukin 2 (IL-2) and interferon gamma (IFNɣ) were measured by commercial ELISA kits (BD Biosciences, Franklin Lakes, NJ, USA).

### 2.8. Construction of AAV Vectors Coding for Nb11

The sequence of a selected Nb able to recognize both mPD-1 and hPD-1 (Nb11) was cloned into a plasmid containing an AAV2 vector backbone downstream different promoters: elongation factor 1α (EF1α), human cytomegalovirus (CMV), spleen focusing-forming virus (SFFV) and the CAG synthetic promoter [27], generating the following plasmids: AAV-EF1α-Nb11, AAV-CMV-Nb11, AAV-SFFV-Nb11, and AAV-CAG-Nb11, respectively. For the construction of these plasmids, a cassette for the expression of Nb11 was synthetized by Integrated DNA Technologies (Coralville, IA, USA), which consisted of: a signal peptide (SP) for secretion in mammalian cells in the amino-terminal end (aminoacidic sequence: NH_2_-MNWGLKLVFFVLILKGVQC-COOH), followed by the optimized Nb11 sequence for its expression in mice, and a HA-tag at the carboxy-terminal end (SP-Nb11-HA cassette), as well as different restriction sites. For the construction of AAV-CMV-Nb11 plasmid, the SP-Nb11-HA cassette was subcloned into pAAV-MCS (Cell Biolabs, San Diego, CA, USA) using Sal I. In the case of AAV-CAG-Nb11, the GFP sequence in AAV-MCS-CBA-GFP plasmid [28] was substituted by the SP-Nb11-HA cassette using Cla I and Xho I. For the construction of pAAV-EF1α-Nb11, SP-Nb11-HA cassette was subcloned into the pAAV-aPDL1 plasmid previously generated by our group [29], using Sal I and Nde I. Finally, the pAAV-SFFV-Nb11 plasmid was generated in two steps: first, EF1α promoter in pAAV-aPDL1 was substituted by the SFFV promoter [30], using EcoR I and Asc I, and second, SP-Nb11-HA cassette was subcloned using Sal I and Nde I, removing the original transgene.

### 2.9. Expression of Nb11 from AAV Vectors In Vitro

Nb11 expression from AAV plasmids was evaluated by transfecting BHK and HuH-7 cell lines using lipofectamine-2000 (Invitrogen, Waltham, MA, USA) in 6-well plates. 48 h later, Nb11 expression was evaluated by immunofluorescence using a mouse anti-HA antibody (BioLegend, San Diego, CA, USA) and an anti-mouse IgG (immunoglobulin G) secondary antibody fused to Alexa-Fluor 488 (Invitrogen, Waltham, MA, USA). Supernatants and cell extracts from transfected cells were collected and evaluated by a binding ELISA against PD-1 as described above, and by Western blot using an anti-HA antibody and an anti-mouse IgG secondary antibody fused to horseradish peroxidase (Sigma, St Louis, MO, USA). Glycosylation of Nb11 was analyzed by Western blot after treatment with the Protein Deglycosylation Mix II (New England Biolabs, Ipswich, MA, USA), according to manufacturer’s instructions. Binding of Nb11 to mPD-1 was also evaluated by immunoprecipitation, using protein G Dynabeads (Invitrogen, Waltham, MA, USA) coated with mPD-1-Fc. Both immunoprecipitated and non-immunoprecipitated fractions were evaluated in Western blot as described above.

A commercial kit was used for the extraction of DNA (Macherey-Nagel, Bethlehem, PA, USA) from transfected cells. Quantitative PCR (qPCR) was performed using primers specific for Nb11 (forward primer: 5′-ACTAGTGCCCAGGTGCAGCTGGTG-3′ and reverse primer: 5′-GGTACCTGAGGAGACAGTGACCTGGGTCC-3′) from DNA, using iQ SYBR Green Supermix and the CFX96 Real-Time Detection System (Bio-Rad, Hercules, CA, USA).

### 2.10. Production of AAV8 Vectors

Recombinant AAV8 viral particles (VPs) were generated by co-transfecting 150 cm^2^ flasks containing confluent HEK-293T cells with each AAV-Nb11 plasmid and pDP8.ape (Plasmid Factory, Bielefeld, Germany), using linear PEI of 25 kDa. After 72 h, cells were collected, treated with lysis buffer (50 mM Tris-Cl, 150 mM NaCl, 2 mM MgCl_2_ and 0.1% Triton X-100) and stored at −80 °C. Supernatants were also collected and treated with PEG-8000 (8% v/v final concentration) for 48–72 h at 4 °C, then they were centrifuged (1500× *g*, 15 min, 4 °C) and the pellet was resuspended in lysis buffer and kept at −80 °C. Samples from cells and supernatants received three cycles of freezing and thawing before purification by ultracentrifugation at 350,000× *g* for 2.5 h in a 15–57% iodioxanol gradient, as described previously [31]. Purified VPs were concentrated using Amicon Ultra Centrifugal Filters-Ultracel 100 K (Millipore, Burlington, MA, USA). For titration, viral genomes (vg) were extracted from DNAse-treated VPs using the High Pure Viral Nucleic Acid Kit (Roche, Basel, Switzerland), and vector titers (vector genomes (vg)/mL) were determined by qPCR using primers specific for Nb11 as described above.

### 2.11. Evaluation of the Safety and Antitumor Activity of AAV8-Nb11 In Vivo

C57BL/6 mice received one dose of 10^11^ vg of the selected AAV-Nb11 vectors in 150 µL by retro-orbital intravenous injection, or 150 µL of saline (control group). Mice were bled at the indicated times. Blood was incubated 30 min at RT to allow coagulation and serum was separated from whole blood by centrifugation at 2300× *g* for 15 min at 4 °C. Serum Nb11 levels were measured by specific hPD-1-Fc binding ELISA assay as described before.

To evaluate toxicity, serum transaminases alanine aminotransferase (ALT) and aspartate aminotransferase (AST) were quantified using a HITACHI C311 analyzer (Roche, Basel, Switzerland), and serum tumor necrosis factor alpha (TNFα) was evaluated with a commercial ELISA kit (BD Biosciences, Franklin Lakes, NJ, USA). Livers were collected from euthanized mice at the end point of the experiment. Formalin-fixed paraffin-embedded liver sections (3 µm thick) were stained with hematoxylin and eosin (H&E) for histological analysis. To evaluate the antitumor activity, mice were challenged subcutaneously with 5 × 10^5^ colon adenocarcinoma MC38 cells in the right flank eight weeks after treatment with AAV-CAG-Nb11 and survival of the animals was followed. Two perpendicular tumor diameters were measured with a caliper every 2–3 days and the product between both measurements was considered as an indicator of tumor size.

### 2.12. Statistical Analysis

All data are expressed as the mean ± SEM. Prism software (GraphPad Software, San Diego, CA, USA) was used for statistical analysis. To compare multiple experimental groups, one-way ANOVA test and Tukey’s multiple comparison test were used, unless indicated otherwise. Unpaired *t* test was used when only two experimental groups were compared. Survival of tumor-bearing mice is represented by Kaplan–Meier plots and analyzed by log-rank test. For time-series analysis (Figures 6a and 7b), data were compared using the extra sum-of-squares F test and fitted to a second-order polynomial equation. The p values < 0.05 were considered statistically significant.

## 3. Results

### 3.1. Generation and Characterization of a Nanobody that Binds to Human and Mouse PD-1

Nbs against PD-1 were generated from a llama immunized with purified recombinant mouse PD-1 (mPD-1) ectodomain (Appendix A) as described in Materials and Methods (M and M). One week after the last immunization, total cDNA was obtained from peripheral blood lymphocytes (PBLs); Nbs genes were amplified with specific primers and cloned into pComb3x phagemid vector as described in Figure 1. In this way, we obtained a phage library of Nbs with a diversity of 2 × 10^8^ clones. The selection of specific Nbs was carried out by three consecutive rounds of bio-panning on HEK-293T cells transiently transfected with a plasmid expressing mPD-1 (Appendix A) as described in M and M. This process allowed us to select a clone (Nb11) with high ability to bind both mouse and human PD-1 in PD-1 binding ELISA assays and by flow-cytometry (Figure 2a–c). Nb11 was subcloned into a pET28-derived vector for its production and purification in *E. coli* as a C-terminally His × 6-tagged protein. Recombinant expression of Nb11 was very efficient and we obtained around 20 mg of purified Nb11 per liter of culture.

The ability of Nb11 to inhibit the PD-1/PD-L1 interaction was evaluated by an ELISA-type assay using the commercially available ectodomains of these proteins as described in M and M. Nb11 was able to inhibit the PD-1/PD-L1 interaction of both mouse and human molecules (Figure 2d,e). In these assays, a Nb with no relevant specificity was included as negative control. The inhibition of mPD-1/mPD-L1 interaction by Nb11 was stronger than the one obtained using a commercial anti-mPD-1 monoclonal antibody (RMP1-14 clone, Bio X Cell, Lebanon, NH, USA), with half maximal inhibitory concentrations (IC50) of 1.7 and 31 nM, respectively (Figure 2d). For hPD-1/hPD-L1 binding inhibition, Nb11 showed an IC50 of 4.3 nM (Figure 2e).

To evaluate the potential biological activity of Nb11, primary T cells isolated from mouse spleens were activated using agonist antibodies against CD3 and CD28, in the presence or absence of mPD-L1-Fc. After 72 h of incubation, levels of IL-2 and IFNɣ in supernatants were measured by ELISA (Figure 3). As expected, the presence of mPD-L1 generated an inhibitory signal that decreased the levels of these activation biomarkers. When Nb11 was added to the culture, the levels of these biomarkers increased, being the effect more evident for IL-2. The same occurred when a blocking mAb against mPD-L1 was used as control. Importantly, a control Nb did not affect these biomarkers, underlining the fact that the effect observed with Nb11 is specific and linked to its blocking capacity. For this experiment, Nbs were previously treated with endotoxin removal columns and absence of endotoxin contamination was confirmed using a commercial kit (Appendix A).

In summary, we describe here the isolation of a new Nb with cross-reactivity for mouse and human PD-1, able to inhibit the binding of PD-1 with PD-L1 of both species in ELISA assays. This Nb has also shown potential biological activity In vitro, reverting mPD-L1-mediated inhibition in mouse T cells.

### 3.2. Nb11 Expression from Mammalian Cells Using AAV Vectors

In this work, an AAV vector was chosen for the delivery of Nb11 gene into cells, due to its good biosafety profile and its ability to achieve long-term transgene expression. Nb11 was subcloned into an AAV2 vector backbone under different promoters, fused to a signal peptide for its secretion and a hemagglutinin (HA) tag for its detection (Figure 4a). Four different strong promoters were compared for the expression of Nb11 from the AAV vector: (i) human elongation factor 1α promoter (EF1α), (ii) CAG promoter, (iii) human cytomegalovirus promoter (CMV) and (iv) spleen focus-forming virus (SFFV) promoter.

Expression of Nb11 from AAV was evaluated in eukaryotic cells in vitro by transfecting plasmids containing these vectors into BHK and HuH-7 cells and analyzing Nb11 expression after 48 h. Transfected cells were first analyzed by immunofluorescence using an anti-HA antibody. We observed a high percentage of cells expressing Nb11 with a bright signal except in the case of Huh-7 transfected with AAV-SFFV-Nb11, where the signal was weaker (Appendix A).

Nb11 levels present in supernatants of transfected cells were quantified by a hPD-1 specific binding ELISA (Figure 4b) and normalized by the amount of transfected DNA (Figure 4c). For normalization, we isolated DNA from transfected cells and analyzed levels of AAV plasmids by qPCR using specific primers for Nb11. AAV-Nb11 DNA levels for the different constructs were very similar, although AAV-CAG-Nb11 showed slightly lower DNA levels, indicating that transfection with this plasmid was less efficient.

The direct ELISA analysis showed that Nb11 expression was significantly lower for the AAV construct having the SFFV promoter in both cell lines, compared to AAV vectors having the other three promoters. Expression levels of Nb11 in supernatants of BHK cells were higher for AAV-CAG-Nb11 (~18 µg/mL), and similar for the AAV vectors harboring EF1α and CMV promoters (~12 µg/mL) (Figure 4b). In the case of HuH-7 cells, expression from AAV-EF1α-Nb11 and AAV-CAG-Nb11 was similar (~4 µg/mL), while AAV-CMV-Nb11 generated lower levels (~1.5 µg/mL). Normalization of ELISA data by Nb11 DNA levels showed very similar results, confirming that AAV-CAG-Nb11 and AAV-SFFV-Nb11 were the constructs that generated the highest and lowest Nb11 levels, respectively (Figure 4c). We also observed that Nb11 was efficiently secreted with all vectors, with more than 90% of total Nb11 present in supernatant, except in the case of AAV-SFFV-Nb11, where percentage of secretion was around 70% (Figure 4d).

Samples from transfected cells were also analyzed by Western blot to confirm the correct expression and secretion of Nb11. In agreement with the ELISA data, cells transfected with AAV-SFFV-Nb11 expressed much lower amounts of Nb11 compared to the other three vectors (Figure 5a). Interestingly, Nb11 expressed from eukaryotic cells showed a double-band pattern that was not observed when expressed from *E. coli* (rNb11). In cell extracts the lower molecular weight (MW) band was stronger, while in supernatants we observed that both bands had similar intensities in BHK cells (Figure 5a) or that the higher MW band was more prevalent in HuH-7 cells (Figure 5b). Reasoning that the higher MW band could be due to glycosylation, supernatant and cell extracts of BHK cells transfected with AAV-EF1α-Nb11 were treated with a mix of glycosidases to remove N- and O-linked glycans. This treatment resulted in the disappearance of the band with higher MW, indicating that in fact Nb11 is glycosylated in mammalian cells (Figure 5c). The higher MW observed for rNb11 can be explained by the additional 6 × His tag, which is absent in Nb11 expressed from AAV vectors.

To analyze whether glycosylation of Nb11 could affect its binding to PD-1, an immunoprecipitation assay was performed using mPD-1-Fc-coated microbeads. Both glycosylated and non-glycosylated Nb11 bands were detected by Western blot after immunoprecipitation with mPD-1-Fc, suggesting that glycosylation does not impair binding of Nb11 to PD-1 (Figure 5d).

### 3.3. AAV Vectors Can Mediate Sustained Nb11 Expression in Mice without Toxicity

To evaluate the expression of Nb11 in vivo, AAV8 viral particles coding for Nb11 were generated from the three AAV vectors that showed higher expression levels in vitro (AAV-CAG-Nb11, AAV-EF1α-Nb11, and AAV-CMV-Nb11). Six-week-old C57BL/6 female mice received one dose of 10^11^ vg (viral genomes) intravenously and the expression of Nb11 was measured in serum by a specific hPD-1 binding ELISA at weeks 0, 1, 2, 4, 8, and 12 post-AAV administration (Figure 6a). In this experiment, performed with a small number of animals (*n* = 3), AAV-CAG-Nb11 showed significantly higher expression of Nb11 in comparison with AAV-CMV-Nb11 and AAV-EF1α-Nb11, reaching the highest levels between weeks 4 and 12, with a peak of 569 ± 307 ng/mL (mean ± SEM) at week 8. In the case of AAV-CMV-Nb11, Nb11 levels increased less pronouncedly until the end of the experiment (week 12), reaching a maximum of 102 ± 38.8 ng/mL. The Nb11 levels obtained with AAV-EF1α-Nb11 were undetectable until week 8, being very low at week 12 (28 ± 21 ng/mL).

To confirm the data obtained with AAV-CAG-Nb11, we evaluated this vector in a larger group of animals (*n* = 12), in which we also assessed toxicity. In this second experiment we observed again a sharp increase of Nb11 expression in serum at four weeks after vector administration, reaching a peak of 2.8 ± 0.4 µg/mL at week six and slightly declining at week eight (Figure 6b). In both experiments, serum Nb11 levels were very low during the first two weeks after AAV administration.

To analyze the safety profile of AAV-CAG-Nb11, we measured different parameters in serum, including inflammation marker TNFα and liver transaminases ALT and AST. Neither TNFα, measured at week six post vector administration, nor transaminases, measured at weeks four and eight, showed differences between vector-treated mice and control mice, indicating that continuous Nb11 expression was safe (Figure 6c). In addition, no significant differences in body weight gain between both groups of animals were observed during the 10-week follow up after treatment (Appendix A). Histological analysis of liver sections at the end point of the experiment showed no liver inflammation in mice that received AAV8-CAG-Nb11 (Appendix A).

### 3.4. AAV-CAG-Nb11 Treatment Can Prevent Tumor Development in Mice

In order to test whether the high Nb11 levels present in the serum of AAV-CAG-Nb11 treated mice could have antitumor potential, we carried out a prophylactic experiment in which mice from the experiment shown in Figure 6b were challenged with MC38 colon adenocarcinoma cells. Tumor cells were injected subcutaneously eight weeks after receiving AAV8-CAG-Nb11, using saline treated mice as control.

Mice expressing Nb11 showed lower frequency of tumor development in comparison with control mice (33% vs. 8%), which was reflected in a significant improvement of survival (Figure 7a). However, although AAV-CAG-Nb11 treatment was successful at preventing tumor development in a fraction of the animals, the overall effect on tumor growth was rather modest (Figure 7b).

## 4. Discussion

Cancer immunotherapy using monoclonal antibodies (mAbs) against immune checkpoints has accomplished remarkable outcomes in the treatment of a variety of solid tumors in the past decades. However, their success is partially limited due to their complex structures and large sizes, leading to high production costs and poor tumor penetration. In contrast, Nbs are nearly ten-times smaller, have simpler structures, show high thermal and physicochemical stability, and are easy to produce in different systems [17]. These characteristics make Nbs very attractive tools for the development of novel therapeutic agents that could replace other formats of antibodies.

Although several Nbs against PD-L1 have already been described [32,33,34], previous work with PD-1 is more limited. Recently, two anti-PD-1 Nbs that inhibit the PD-1/PD-L1 interaction have been reported, against human [35] and mouse PD-1 [36], although none of them have been tested in vivo. In this study, we have identified a new Nb that recognizes PD-1 from both human and mouse origin, named Nb11, and we have demonstrated that it is able to inhibit the binding of PD-1 with its main ligand PD-L1 for both species In vitro. Cross-reactivity is a very desirable feature in translational research, and, to our knowledge, this is the first time that it has been reported for a PD-1-specific Nb.

Expression of Nb11 from *E. coli* was extremely efficient, and we were able to obtain approximately 20 mg of purified Nb11 per liter of culture. This makes Nb11 an inexpensive tool that could be easily used in research or diagnostic applications when expressed recombinantly from bacteria.

Although Nbs have great potential as theranostic agents, their small size also comes with a downside: they are eliminated from bloodstream very quickly. To overcome this disadvantage, an array of different strategies can be employed [37]. In this study, we propose that sustained expression of the therapeutic Nb using gene therapy may compensate its fast clearance in vivo. Several Nbs have already been tested in vivo using gene therapy vectors, mainly adenoviral vectors for passive immunization [38,39,40]. Here, an AAV-based viral vector was chosen for expression of Nb11 because this type of vector has shown a good safety profile, it is easy to produce at high scale and it enables long-term expression of the desired transgene [41]. In addition, AAV vectors expressing antitumoral cytokines, such as IL-12 [42] or IL-27 [43] have been successfully used in tumor preclinical models. Some groups have also used AAV vectors to express Nbs. Verhelle et al. have used AAV9 to express a bispecific Nb to treat amyloidosis in a mouse model of this disease [44], and Del Rosario et al. have used AAV8 to express an influenza virus neutralizing Nb fused to an Fc domain to protect mice from viral infection [45].

In this study, we compared four different strong promoters for the expression of Nb11 from AAV. In vitro, Nb11 was expressed at high levels and secreted very efficiently from three of the evaluated promoters (EF1α, CAG, and CMV) in both BHK and HuH-7 cells. Interestingly, Nb11 levels produced in vitro by these vectors were approximately 100-fold higher than the ones obtained for an anti-PD-L1 mAb expressed from an AAV vector having the EF1α promoter recently reported by us [29]. Unexpectedly, expression from SFFV promoter was significantly lower than from the other three tested promoters. This observation, which contrasts to previous reports in which the SFFV promoter had shown to be superior or equal to the other tested promoters [46,47], suggest that Nb expression in mammalian cells could be greatly affected by the promoter selection. Evaluation of Nb11 expressed from mammalian cells led to the observation that a considerable fraction of the secreted Nb was glycosylated, although this did not seem to affect the ability of Nb11 to bind to its antigen.

In vivo, AAV-CAG-Nb11 was the vector that generated the highest Nb11 expression, which also matched with its higher expression observed in vitro (Figure 4d). However, serum levels of Nb11 increased very slowly, being very low during the first two weeks and reaching the highest levels between weeks 4 and 8 after AAV administration. In preclinical tumor models where tumor development is fast, this delay in Nb11 expression may be a disadvantage, especially if these vectors are used for therapeutic treatment. In this type of approach, Nb11 therapeutic levels could be reached only when tumors are probably too large to be eliminated. In contrast, tumor growth rate in patients is usually much slower than in mouse transplantable tumor models, which would probably make this delay in Nb expression a less crucial concern. Taking into account the limitations of preclinical models, we carried out a proof-of-concept experiment in which we tested the antitumoral effect of Nb11 by evaluating its protective potential against MC38 tumor challenge, in mice expressing high levels of Nb11 in serum. In this experiment, we observed a higher protection against tumor development in mice treated with AAV-CAG-Nb11 compared to mice treated with saline (33% vs. 8%), which led to a significantly higher survival rate for the group expressing Nb11. This observation indicates that Nb11 has antitumor activity in the MC38 tumor model. We and others have previously shown that MC38 cells express PD-L1 in vivo, which can probably help these tumors escape immune responses [29]. The presence of an anti-PD-1 Nb in mice having MC38 tumors could probably help reestablish these antitumor responses by blocking PD-L1 interaction with PD-1 expressed by T lymphocytes.

Importantly, long-term expression of Nb11 in vivo did not show evidence of toxicity in mice, indicating that it could represent a safe therapeutic agent. To evaluate toxicity, we focused on analyzing potential hepatotoxicity by measuring serum transaminases and liver histology, as AAV8 vector has a very strong tropism for liver [48]. Neither systemic inflammation, nor liver toxicity, nor delay in body weight gain were observed in treated animals compared to saline. Although mice do not usually show toxicity when treated with mAbs against PD-1, liver injury has been reported when PD-1 blockade was combined with cytotoxic T lymphocyte antigen 4 (CTLA-4) and indoleamine 2,3-dioxygenase 1 (IDO1) blockers [49]. Although we have only used one checkpoint blocker, its continuous expression in the liver did not seem to induce hepatic toxicity.

As the turnover of adult hepatocytes is very slow, the aim of using AAV8 vector is to achieve, with a single administration, stable transgene expression for a long period of time. Thus, the use of AAV vectors to express immunomodulatory mAbs in vivo may lead to improved outcomes and reduced adverse effects over conventional protein administration, as it would avoid the need for administration of high doses of mAbs systemically [50,51]. In order to optimize this type of gene therapy, thus increasing the bioavailability of the therapeutic Nb in the tumor, minimizing off-target toxicity, and inhibiting its expression when necessary, different strategies may be pursued in the future: (1) retargeting AAV vector to the desired tissue by modifying capsid proteins [52,53]; (2) directing the therapeutic Nb to the tumor, by fusion to a tumor-targeting Nb or peptide [47,54,55]; and (3) including tissue-specific promoters and inducible expression systems to be able to shut off expression when desired [42,50,51]. Moreover, in foreseeing clinical application, humanization of Nb11 should be considered to minimize its potential immunogenicity [56].

In summary, we describe in this study the identification and characterization of a novel PD-1-specific Nb able to block PD-1/PD-L1 interactions for both murine and human molecules. Our results show that expression of this Nb from AAV vectors in mice can be detected for at least 12-weeks of follow-up, with no evidence of toxicity. As prophylactic treatment, AAV-CAG-Nb11 conferred protection to a significant fraction of animals from MC38 tumor challenge, indicating that Nb11 may be an appealing therapeutic agent for cancer immunotherapy.

## Figures and Tables

**Figure 1 biomedicines-08-00562-f001:**
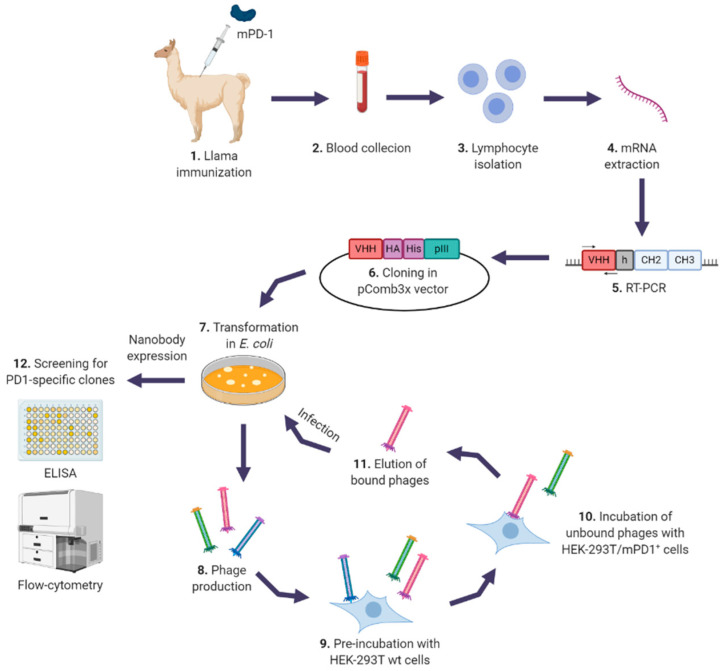
Construction of anti-programmed death-1 (PD-1) nanobodies (Nbs) phage-display library. Schematic representation of the generation and selection of PD-1 specific Nbs from an immunized llama. After immunization (**1**), collection of blood and extraction of mRNA from peripheral lymphocytes was performed (**2–4**). VHH (Nbs) genes were amplified with specific primers (**5**) and cloned into pComb3x phagemid vector (**6**). The library was electroporated into *E. coli* (**7**), phages were produced (**8**) and used for the selection of PD-1 specific Nbs using HEK-293T cells transiently expressing mPD-1 (HEK-293T/mPD-1^+^ cells) (**10**), after a preincubation with HEK-293T wild type (wt) cells (**9**). Eluted phages (**11**) were used to infect *E. coli* to perform another round of selection. After three rounds, screening of positive clones was performed in PD-1 binding ELISAs and flow-cytometry assays, using HEK-293T cells expressing mouse or human PD-1 (**12**). Created with BioRender.com.

**Figure 2 biomedicines-08-00562-f002:**
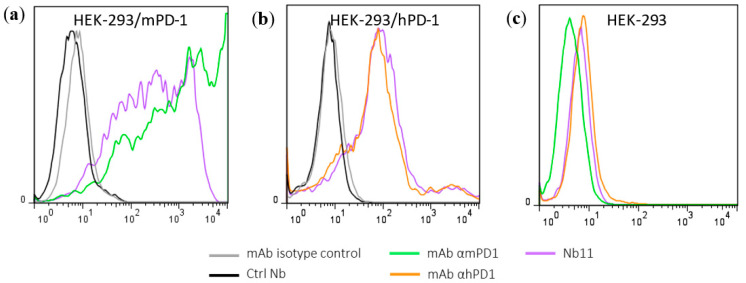
Identification of a PD-1 specific Nb able to inhibit PD-1/PD-L1 interaction. Evaluation of Nb11 reactivity by flow-cytometry using HEK-293T cells transiently transfected with (**a**) mouse PD-1 (mPD-1), (**b**) human PD-1 (hPD-1) or (**c**) untransfected. Commercial monoclonal antibodies (mAbs) against mPD-1 and hPD-1 were included in this analysis, as well as their corresponding isotype controls and a control Nb (Ctrl Nb) with no relevant specificity. Inhibition curves of PD-1/PD-L1 interaction using Nb11, for (**d**) mouse and (**e**) human molecules. In (**d**), ELISA plates were coated with mPD-L1-Fc and incubated with biotinylated mPD-1-Fc in the presence of different concentrations of Nb11, or of the indicated control molecules. In (**e**), ELISA plates were coated with hPD-1-Fc and incubated with Nb11 at different concentrations together with biotinylated hPD-L1-Fc. In this case, hPD-L1-Fc without biotin (soluble hPD-L1-Fc) was used as inhibition control. PD-1/PD-L1 binding was detected with streptavidin-peroxidase. Data represent mean ± SD.

**Figure 3 biomedicines-08-00562-f003:**
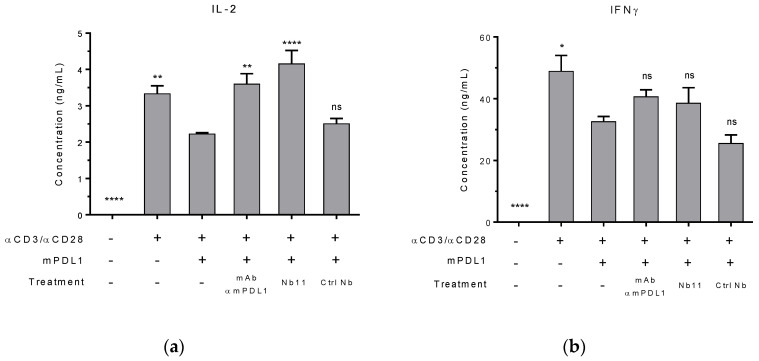
Nb11-mediated restoration of activity markers in mouse primary T cells. T cells were activated with antibodies against CD3 and CD28, in the presence or absence of mPD-L1. Cells were incubated with Nb11, an anti-mPD-L1 control mAb, a control Nb (Ctrl Nb) or without antibodies. After 72 h of incubation, supernatants were collected and levels of (**a**) IL-2 (interleukin 2) and (**b**) IFNɣ (interferon gamma) were measured by ELISA. Data represent the mean + SEM. Statistical analysis was performed using one-way ANOVA test with Dunnett’s multiple comparison test, using as reference group the third column (activation with antibodies and mPD-L1, without treatment). One representative experiment out of two performed is shown. * *p* < 0.05; ** *p* < 0.01; **** *p* < 0.0001, ns: not significant.

**Figure 4 biomedicines-08-00562-f004:**
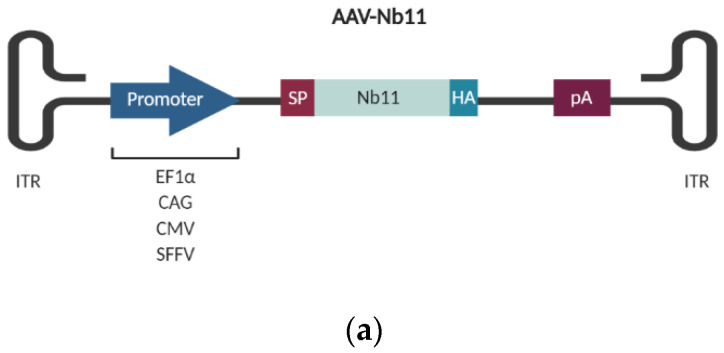
Quantification of Nb11 expressed from BHK and HuH-7 eukaryotic cell lines. Cells were transfected with 2 µg/well of adeno-associated virus (AAV) plasmids coding for Nb11 under the indicated promoters, and 48 h later supernatants and cell extracts were collected for analysis. (**a**) Schematic representation of AAV vectors coding for Nb11 under the control of four different promoters. (**b**) Nb11 levels in supernatants of transfected BHK and HuH-7 cells, measured by a hPD-1 binding ELISA. Data represent the mean + SEM. (**c**) Relative levels of Nb11 in supernatants normalized with Nb11 DNA levels quantified from transfected cells, expressed as Nb11 concentration (ng/mL)/2^ΔCt^. (**d**) Analysis of the secretion of Nb11 in transfected BHK and HuH-7 cells. The percentage of secretion was calculated by quantifying the total amount of Nb11 present in supernatants and cell lysates and determining the percentage of Nb11 present in each type of sample. Asterisks above bars indicate comparison of each group with mock. Other comparisons are indicated by horizontal bars. * *p* < 0.05, ** *p* < 0.01, *** *p* <0.001, **** *p* < 0.0001, ns: not significant. ITR: AAV inverted terminal repeats; EF1α: human elongation factor 1α promoter; CMV: human cytomegalovirus promoter; SFFV: Spleen focus-forming virus promoter; SP: signal peptide; HA: hemagglutinin tag; pA: synthetic polyadenylation sequence.

**Figure 5 biomedicines-08-00562-f005:**
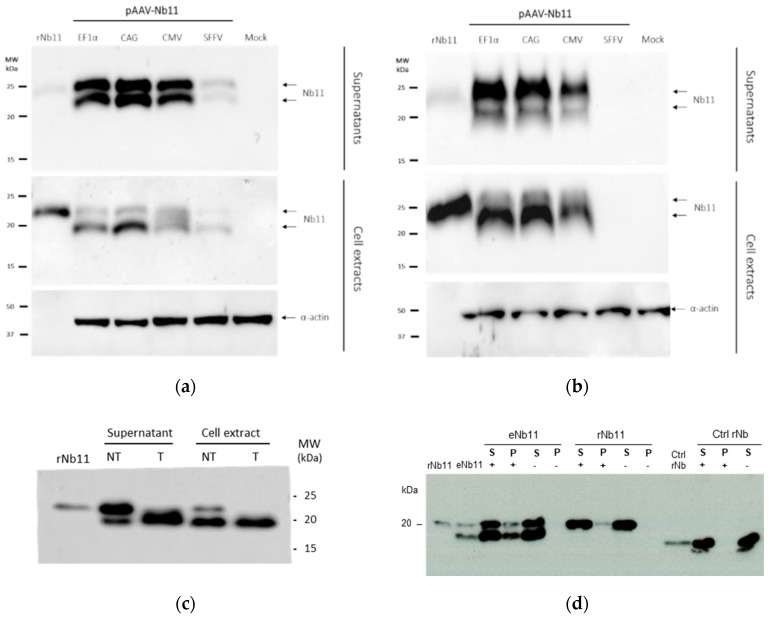
Western blot analysis of Nb11 expressed from eukaryotic cells In vitro. Supernatant (S) and cell extracts from (**a**) BHK and (**b**) HuH-7 cells transfected with AAV plasmids coding for Nb11, under the promoters indicated above the gels, were analyzed by Western blot using an anti-HA antibody. Nb11 in HuH-7 cell samples transfected with pAAV-SFFV-Nb11 could not be detected in this analysis. Twenty nanograms of rNb11 (produced in bacteria) were loaded as control. (**c**) Analysis of glycosylation in supernatant and cell extracts from BHK cells transfected with pAAV-EF1α-Nb11 that were treated (T) or non-treated (NT) with a mix of N-and O-glycosidases. (**d**) Immunoprecipitation of Nb11 using mPD-1. Supernatants from BHK cells transfected with pAAV-EF1α-Nb11 (eNb11) or rNb11 were incubated with microbeads coated (+) or uncoated (−) with mPD-1-Fc. After incubation, microbeads were magnetized and the mPD-1-bound fraction (P: precipitated) or unbound fraction (S: soluble) were separated for Western blot analysis using an anti-HA antibody. Ctrl rNb, recombinant control Nb having an HA tag with no relevant specificity.

**Figure 6 biomedicines-08-00562-f006:**
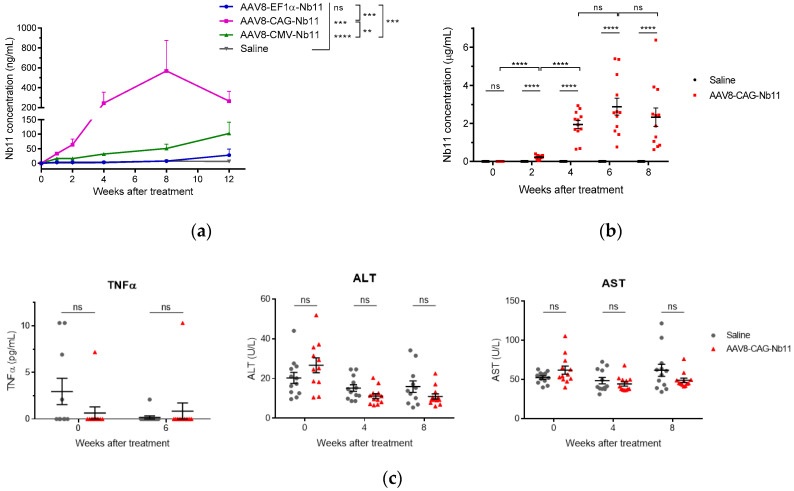
Analysis of Nb11 expression and toxicity in mice treated with AAV vectors. C57BL/6 mice received one dose of the indicated AAV-Nb11 vectors systemically (10^11^ vg/mouse), and expression of Nb11 was analyzed over time by measuring its levels in serum with a hPD-1 binding ELISA. (**a**) Serum levels of Nb11 in mice that received AAV8 particles coding for Nb11 under different promoters (*n* = 3). Data represent mean + SEM. (**b**) Serum levels of Nb11 in mice treated with AAV-CAG-C11 or saline (*n* = 12). (**c**) Serum levels of TNFα, AST, and ALT in mice treated with AAV-CAG-C11 or saline. Data represent mean ± SEM. ** *p* < 0.01, *** *p* < 0.001, **** *p* < 0.0001; ns: not significant.

**Figure 7 biomedicines-08-00562-f007:**
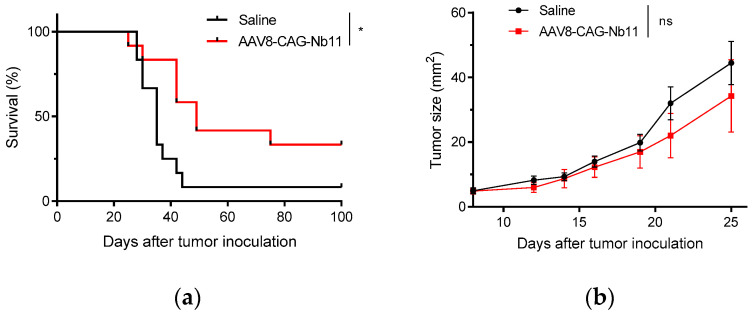
Prevention of tumor development in mice treated with AAV-CAG-Nb11. C57BL/6 mice that had received 10^11^ vg of AAV-CAG-Nb11, and control mice (that received saline solution), were challenged with 5 × 10^5^ MC38 cells eight weeks after treatment. (**a**) Survival after tumor challenge. (**b**) Mean tumor size evolution (mm^2^) ± SEM. * *p* < 0.05, ns: not significant.

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
