# Peer review of "Long-Term Systemic Expression of a Novel PD-1 Blocking Nanobody from an AAV Vector Provides Antitumor Activity without Toxicity"

_biomedicines, 2020, doi:10.3390/biomedicines8120562_

Round 1
Reviewer 1 Report
In this manuscript, Silva-Pilipich et al. describe a study in which they generated a novel nanobody against immune checkpoint molecule PD-1 and confirmed that this nanobody has antitumor activity with low cytotoxicity. As a new generation of a therapeutic antibody, nanobody has many advantages, including tiny molecule mass, high stability, more specificity, and great binding affinity. In this study, the authors did abundant experiments to demonstrate the generated nanobody Nb11 can bind human and mouse PD-1. Through this mechanism, Nb11 could restore the mouse T cell activity and prevent tumor progression in a mouse model with low toxicity. This study is quite convincing and provides very useful information for nanobody-based immunotherapy.
Some minor comments are listed below.
- I don’t think that the E. coli-producing nanobody has a similar efficacy of Ag binding and tumor restriction with the mammalian cell-producing nanobody.
- Did the authors compare the antitumor activity of commercial anti-PD1 antibodies (e.g. Nivolumab and Pembrolizumab) with that of Nb11?
- The antitumor activity of Nb11 is just modest in the mouse model. Did the authors try to give more doses of Nb11 to increase its antitumor efficiency since it is low toxicity?
Author Response
Please see attachment。

Reviewer 2 Report
Line 295, quality of purified mPD-1 can be provided as SI.
Line 300, quality of HEK293 cells displaying mPD-1 can be provided as SI.
Line 304, endotoxin removal results should be provided as SI.
Fig 2a, the broad ranges on fluorescent signals for Nb11 and anti-mPD1 need to be explained.
How such phenomena happened and why not for the results with HEK cells displaying hPD1 (Fig 2b)?
Line 317, “as described in M&M” for non-standard experiment protocol (study specific ones), it will be better having a brief description in Results
In vivo test with mice by AAV administration, what could be the tissues / cells express Nb11? Such discussion will be beneficial.
Nb11 was detectable 12 week after AAV administration, will Nb11 will be permanent? What will be the fate of introduced AAV and its encoding gene? These potential issues associated with gene therapy are worth for discussion.
Round 2
Reviewer 1 Report
All of the questions have been answered adequately. It has reached the level of publication.
Reviewer 2 Report
All my review comments are addressed.